# Significance of Autoantibodies to Ki/SL as Biomarkers for Systemic Lupus Erythematosus and Sicca Syndrome

**DOI:** 10.3390/jcm11123529

**Published:** 2022-06-20

**Authors:** Michael Mahler, Chelsea Bentow, Mary-Ann Aure, Marvin J. Fritzler, Minoru Satoh

**Affiliations:** 1Werfen Autoimmunity, San Diego, CA 92131, USA; cbentow@werfen.com (C.B.); maure@werfen.com (M.-A.A.); 2Department of Medicine, Cumming School of Medicine, University of Calgary, Calgary, AB T2N 1N4, Canada; fritzler@ucalgary.ca; 3Department of Clinical Nursing, School of Health Sciences, University of Occupational and Environmental Health, Kitakyushu 807-8555, Japan; satohm@health.uoeh-u.ac.jp

**Keywords:** Ki/SL, proteasome, autoantibodies, lupus, SLE, Sjögren syndrome

## Abstract

Anti-Ki/SL antibodies were first described in 1981 and have been associated with systemic lupus erythematosus (SLE) and Sicca syndrome. Despite the long history, very little is known about this autoantibody system, and significant confusion persists. Anti-Ki/SL antibodies target a 32 kDa protein (also known as PSME3, HEL-S-283, PA28ƴ, REGƴ, proteasome activator subunit 3), which is part of the proteasome complex. Depending on the assay used and the cohort studied, the antibodies have been reported in approximately 20% of SLE patients with high disease specificity as compared to non-connective tissue disease controls. The aim of this review is to summarize the history and key publications, and to explore future direction of anti-Ki/SL antibodies.

## 1. Introduction

Although known for more than four decades (Figure 1), very few details are known about anti-Ki/SL antibodies, and confusion persists. Historically, the nomenclature of the Ki/SL target antigen included SL (Sicca Lupus), PL-2 and Ki [1,2]. In addition, several other names can be found, including PSME3, HEL-S-283, PA28ƴ, REGƴ, proteasome activator subunit 3. Eventually, it was concluded that this was indeed a single autoantibody system, now named Ki/SL. When anti-Ki antibodies were first described by Tojo et al. [3], and almost in parallel by Harmon et al. [4], as was the convention at the time, Tojo et al. named the novel autoantibody after the index patient Kikuta (Ki) [3], and Harmon et al. [4] choose to link it to the clinical association Sicca/lupus (SL). Early evidence using double immunodiffusion showed that they were identified in approximately 10% of SLE sera and were often associated with anti-Sm autoantibodies.

Initially, some sources confused the Ki with Ku/DNA-PKcs (DNA-dependent phosphokinase catalytic subunit) [5], but it was clearly demonstrated that anti-Ki/SL autoantibodies recognize a 32 kDa protein, a soluble subunit of the nuclear PA-28 (proteasome activator) protein family, which is unrelated to the Ku/DNA-PKcs antigens [2]. The confusion from the study by Francoeur et al. [5] arose because the serum that Tojo sent to Francoeur contained both anti-Ku and Ki/SL antibodies. Due to the strong presence of Ku-specific bands in immunoprecipitation (IP), the 32 kDa protein band was overlooked, and it was concluded that anti-Ki and anti-Ku were identical. Unlike other systemic lupus erythematosus (SLE)-related autoantigens, such as Sm and U1RNP, Ki/SL was not associated with detectable RNA species [2]. Some studies focused on another autoantibody system in SLE termed Ki-67, which added to the confusion [6].

## 2. Materials and Methods

Due to the limited number of studies and the heterogenicity of methods and observations, our aim was to summarize the current knowledge in a narrative review using the search terms (Ki+ autoantibodies; SL+ autoantibodies, Ki/SL+ antibodies) instead of a systematic literature review.

## 3. Clinical and Demographic Association of Anti-Ki/SL Antibodies

Although there are no meta-data available as of today, mostly due to the limited number of studies and the heterogenicity of the methods used to detect anti-Ki/SL antibodies, we concluded that anti-Ki/SL antibodies are mostly found in SLE patients followed by patients with Sjögren syndrome (SjS) or Sicca syndrome [7,8], depending on the clinical definition. Especially in SLE, autoantibodies to a wide range of antigens have been reported, and anti-Ki/SL is part of the ever-expanding list [9]. High prevalence of anti-Ki/SL antibodies was also observed in patients with the overlap syndrome [3] and systemic sclerosis (SSc) [8,10]; however, in these studies, the number of patients was relatively small. In one of the earliest and largest clinical and serological studies of 516 connective tissue disease (CTD) patients, anti-Ki/SL autoantibodies were found in 12% of SLE patients, 14% of patients with mixed connective tissue disease (MCTD), 18% of patients with vasculopathies and 3% of patients with SjS [11]. Early clinical correlation studies focused on SLE patients indicated that anti-Ki/SL autoantibodies were associated with malar rash and multiple ANA specificities [7]. Another report of clinical, serological and HLA data from 119 SLE patients found no clear clinical associations with anti-Ki/SL antibodies, except for a higher frequency of non-infective fever [12], Sicca syndrome and skin involvement [13]. Fredi et al. [14] focused on anti-Ki-SL antibodies in SLE patients and reported, based on multivariate analysis, that anti-Ki/SL was significantly associated with male sex (*p* = 0.017), an observation, which is in line with the early work by Riboldi et al. [11], Cavazzana et al. [7] and Fredi et al. [14]. Although no systematic study has been conducted until today, it appears that anti-Ki/SL antibodies can be found in patients with a wide range of ethnicities [15].

When more sensitive ELISA methods, using purified native Ki/SL antigens, were used to analyze the clinical and serologic features of SLE, a higher prevalence of central nervous system involvement was noted [10]. Outside SLE and other CTD, anti-proteasome antibodies have been studied in psoriasis patients [16].

## 4. Case Reports and Longitudinal Analysis of anti-Ki/SL Antibodies

Several case reports have been published on patients exhibiting anti-Ki/SL antibodies [13,17,18,19,20], including a patient with fatal CTD overlap syndrome [13], a patient with SSc/dermatomyositis (DM) overlap syndrome, an individual with anti-centromere positive pulmonary-renal syndrome [18], a case with SLE with epileptic seizures and chorea during prednisolone treatment [16], an individual with SSc with interstitial pneumonia and various autoantibodies (improvement by intravenous cyclophosphamide therapy) [20] (Table 1). In addition to the studies measuring anti-Ki/SL antibodies during a single timepoint (mostly at diagnosis), one case report also provided longitudinal analysis. In this case of a female SLE patient, the titer of anti-Ki/SL antibody rose before the onset of pericarditis and pleuritis, suggesting that anti-Ki/SL titers might reflect disease activity [8]. Although case reports and case series do not allow us to draw strong conclusions about clinical utility, they provide valuable reference points for future studies.

## 5. Epitope Distribution on the Proteasome Complex and on Ki/SL

Ki/SL is part of the human proteasome macromolecular complex, which is a known target of several autoantibodies [2,21,22,23,24,25]. Studies aimed to identify the reactive epitope of autoantibodies on the Ki/SL antigen [26,27,28]. Using different methods, including recombinant protein fragments and synthetic peptides, multiple epitopes were mapped to different regions of the protein (see Figure 2) that were associated with distinctive immune responses and certain clinical subtypes [5,15]. Interestingly, a short peptide sequence (named KILT) was identified [26,28], which bound antibodies in 18/49 (36.7%) anti-Ki/SL positive serum samples. A preliminary analysis indicates that KILT exhibited different clinical associations when compared to the full-length protein, a finding that needs to be validated in larger cohorts. Similarly, patients with antibodies that react with both N- and C-terminal areas are reported to have higher prevalence of the Sicca syndrome [27].

## 6. Detection Methods for Anti-Ki/SL Antibodies

### 6.1. Indirect Immunofluorescence Pattern of Anti-Ki/SL Antibodies

The characteristic indirect immunofluorescent (IIF) staining pattern of anti-Ki/SL antibodies was reported to be diffuse speckled nuclear on HEp-2 cells, although some substrates showed nucleolar staining as well [29] (Figure 3). Interestingly, antibodies to PA28a showed cytoplasmic staining, which is consistent with the reported localization of the protein and also with the moderate (~40%) homology between Ki/SL and PA28a, as the cognate antibodies are apparently not cross-reactive [30]. More specifically, although 13/27 (48%) of anti-Ki/SL also reacted with PA28a, it is unlikely that this represents cross-reactivity. Until the present, only one study that investigated the reactivity of anti-PA28a and anti-Ki/SL in the same cohort of patients [30] found that the prevalence of the two autoantibodies was comparable. Anti-Ki/SL antibodies have not been addressed by the International Consensus of ANA Patterns (ICAP) [31]; however, the described pattern is similar to AC-04 and/or AC-05. Along those lines, it is of relevance that more and more sub-patterns are being added to the consensus list [32]. Interestingly, anti-Ki/SL antibodies frequently occur at high titers, both using IIF as well as solid-phase assays, such as ELISA (unpublished data).

### 6.2. Other Detection Methods for Anti-Ki/SL Antibodies

Historically, anti-Ki/SL antibodies were initially detected by double immunodiffusion (DID) and IP [5]. The first ELISA was based on a native Ki/SL antigen purified from rabbit thymus by ammonium sulfate precipitation and affinity chromatography, followed by high-pressure liquid chromatography gel filtration [10]. In total, 30 out of 140 (21.4%) patients with SLE had anti-Ki/SL antibody by ELISA, whereas 11 (7.9%) were positive by DID. In the early 1990s, when an ELISA system utilizing a recombinant human protein was used to test samples from 220 patients with various CTDs, anti-Ki/SL antibodies were detected in 18.9% of SLE sera [8]. Consequently, the method rather than the source of antigen (recombinant *vs.* native) affects the prevalence of the antibodies in disease cohorts.

## 7. Co-Expression of Anti-Ki/SL and Other Autoantibodies

Anti-Ki/SL antibodies have been associated with several other autoantibodies, including anti-Sm [5], anti-Ro [2], anti-Ku, as well as anti-proliferating cell nuclear antigen (PCNA) [2,11] (Table 2). However, no clear consensus has been established, as some studies resulted in conflicting findings. As an example, a study by Fredi et al. [14] identified anti-Ki/SL antibodies in 31 patients, of which about one-half had no accompanying antibodies.

## 8. Future Directions

Future studies should re-evaluate the serological and clinical associations of anti-Ki/SL antibodies and also include experiments to shed more light on the potential associations with disease activity and treatment response in SLE patients. Along those lines, it is noteworthy that protease inhibitors have shown promise in treatment of refractory SLE [33,34]. Whether this is related to the proteasome levels or activity in serum or with the presence of anti-Ki/SL antibodies is a matter of future studies. Ideally, such investigations of the clinical phenotypes should be performed on inception cohorts of SLE patients, such as the SLICC cohort [35]. Lastly, with the intent to identify pre-clinical autoimmune conditions (e.g., early SLE), studies of cohorts, such as the US military, might provide valuable insights [36].

## Figures and Tables

**Figure 1 jcm-11-03529-f001:**
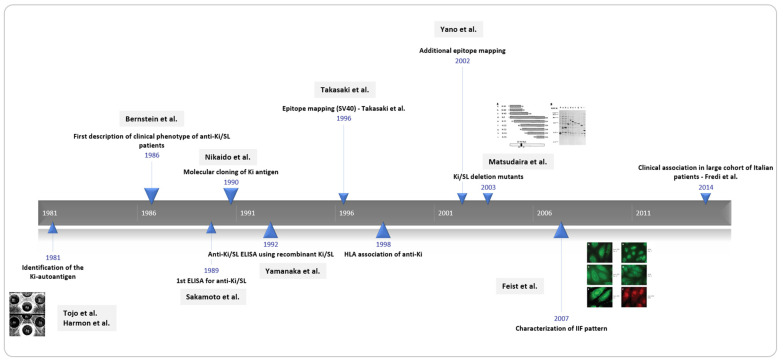
Four historical decades of anti-Ki/SL antibodies. The history of anti-Ki/SL antibodies started with the discovery by Tojo and Harmon et al. in 1981, followed by several clinical association and epitope mapping studies. ELISA = enzyme linked immunoassay; HLA = Human Leukocyte Antigen; SL=sicca lupus.

**Figure 2 jcm-11-03529-f002:**
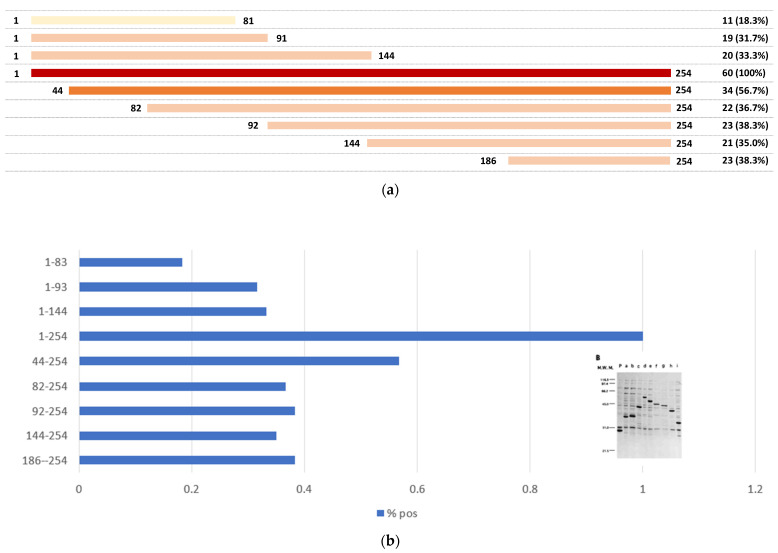
Epitope distribution on the Ki/SL antigen. (**a**) shows a visual representation of the recombinant truncated fragments (and full-length) of the Ki/SL antigen and the corresponding reactivity study by Matsudaira et al. [27] (Panel (**b**)) Shows the fraction of patients reacting with the recombinant fragments.

**Figure 3 jcm-11-03529-f003:**
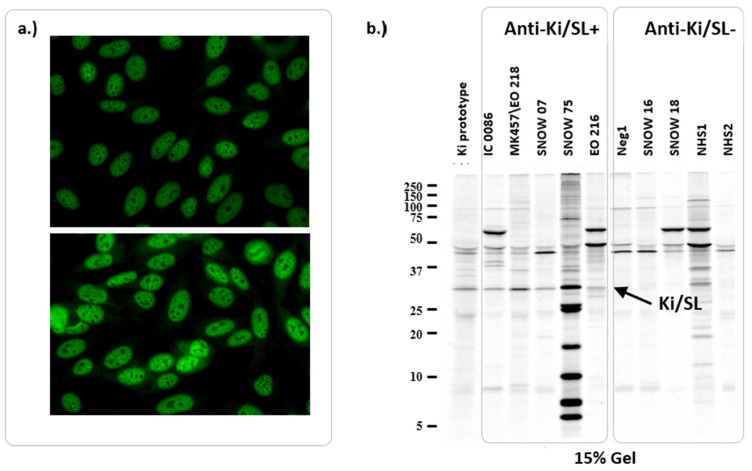
Detection methods for anti-Ki/SL antibodies. (**a**). Indirect immunofluorescence patterns on HEp-2 slides showing a nuclear speckled pattern. (**b**). Immunoprecipitation pattern shows the immunoprecipitation (IP) bands associated with the presence of anti-Ki/SL antibodies.

**Table 1 jcm-11-03529-t001:** Overview of case studies including the measurement of anti-Ki/SL antibodies.

Case Study	Diagnosis	Comments	Ref
Ishiyama 1996	SSc/ILD	-	[20]
Wakasugi 1996	SLE/epileptic seizure/chorea	-	[19]
Oide 2001	Pulmonary-renal syndrome	-	[18]
Miyachi 2002	SSc/DM overlap	anti-Ku and anti-Ki/SL	[17]

DM = dermatomyositis; ILD = interstitial lung disease; SLE = systemic lupus erythematosus; SSc = systemic sclerosis.

**Table 2 jcm-11-03529-t002:** Prevalence of anti-Ki/SL antibodies in different diseases.

Disease	Tojo et al. 1981	Bernstein et al. 1986	Riboldi et al. 1987	Boey et al. 1988	Sakamoto et al. 1989	Yamanaka et al. 1992	Fredi et al. 2014
**Method**	**DID**	**CIE**	**CIE**	**DID**	**ELISA**	**ELISA**	**CIE**
SLE	30/255 (11.8%)	20/300 (6.7%)	27/217 (12.4%)	8/94 (8.5%)	30/140 (21.4%)	21/111 (18.9%)	31/540 (5.8%)
SjS			1/38 (2.6%)			2/25 (8.0%)	
SS		2/60 (3.3%)					
SSc	0/90 (0.0%)		0/119 (0.0%)		3/25 (12.0%)	2/30 (6.7%)	
PM/DM	0/29 (0.0%)		0/14 (0.0%)		(0.0%)	1/30 (3.3%)	
RA	0/33 (0.0%)	2/70 (2.9%)	0/37 (0.0%)		(1.4%)	2/50 (4.0%)	
OS	7/36 (19.4%)						
PN	0/6 (0.0%)						
MCTD		1/50 (2.0%)	3/21 (14.3%)		1/12 (8.3%)		
HI			0/28 (0.0%)		(0.0%)		
PBC		1/135 (0.7%)					
ITP		1/110 (0.9%)					
VAS			2/11 (18.2%)				
pRP			0/59 (0.0%)				
**Demographics**							
Male sex			yes				yes
Other associations	Arthritis/pericarditis, Sm	White SLE, Ro(SS-A), PCNA	PCNA		CNS, Sm		

Abbreviations: DM, dermatomyositis; HI, healthy individuals; ITP, idiopathic thrombocytopenic purpura; MCTD, mixed connective tissue disease; OS, overlap syndrome; PBC, primary biliary cholangitis; PM, polymyositis; PN, periarteritis nodosa; SS, Sicca syndrome; RA, rheumatoid arthritis; SjS, Sjögren’s syndrome; SLE, systemic lupus erythematosus; SSc, systemic sclerosis.

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
