# Peer review of "Significance of Autoantibodies to Ki/SL as Biomarkers for Systemic Lupus Erythematosus and Sicca Syndrome"

_jcm, 2022, doi:10.3390/jcm11123529_

Round 1

Reviewer 1 Report

This is an interesting narrative review of anti-Ki/SL, a potentially forgotten autoantibody. Since 1981, after the discovery of anti-Ki/SL, it has been found in several autoimmune diseases, most notably SLE. In this context, no association was found with a particular subset of the disease except, according to some authors, the higher prevalence in the male gender.

In this manuscript the authors have described the different methods used to detect anti-Ki/SL. Unfortunately, a reliable commercial laboratory test is not currently available. This is the main reason why it is currently almost impossible to test it in routine laboratories. However, in the past, even using relatively insensitive methods such as DID or CIE, anti-Ki/SL was detectable in a considerable number of SLE patients. Although several combinations of antibodies have been proposed, in about half of SLE cases it was detected in isolation.

I have no relevant comments to make.

I only recommend a thorough proofreading review to correct the numerous typos, including the PCNA spelling mistake in the last line of Table 2.

Author Response

Thank you for your review and the feedback. We reviewed the manuscript carefully and made several corrections:

Abstract line 13

PA28-gamma, PA28G, --- PA28g (symbol)

P1 line23

Know --- known

P1 line26

PA28-gamma, PA28G, --- PA28g (symbol)

P6 line 137

Wasused --- was used

P6 line 139

Rather that --- rather than

P7 Table 2

Column Bernstein, Other association

PNCA --- PCNA

Column Riboldi, other association

PNCA --- PCNA

P6 Fig3A

Is it possible to replace with images of smaller number of cells at higher magnification?

Fig3B

number of lanes (n=12) and labeling (n=11) do not match and not well aligned.

Remove the U907 lane if possible.

Molecular weight markers are 250, 150, 100, 75, 50, 37, 25, 20, 10, 5kD

15% gel ---- 13% SDS-PAGE

Reviewer 2 Report

The manuscript is well organized and I think it is an excellent review for Ki/SL antibody.

Author Response

Thank you for reviewing our manuscript on anti-Ki/SL antibodies

Reviewer 3 Report

Dear authors

This is a review article focusing on the significance of autoantibodies against Ki/SL. Historical, clinical and case reports data are being presented and discussed. In addition, the detection methods, the relevant epitopes, the co-existence with other autoantibodies and future direction are also discussed. However, the following comments should be considered:

  • Anti-Ki/SL autoantibody is one of the many experimental autoantibodies that have been described in the literature of systemic autoimmune diseases and especially lupus, but has not drawn any clinical attention.
  • The described clinical associations of anti-Ki/SL autoantibodies are non-specific, ranging from lupus to vasculopathies.
  • The reported clinical associations in lupus patients include malar rah, sicca and male sex which are of no clinical importance.
  • Anti-Ki/SL autoantibodies have no special features and data are sparse to justify a review article.

Author Response

Thank you for your feedback on our review. We agree that the knowledge on anti-Ki/SL is limited. One of the reasons why anti-Ki/SL antibodies are not studied is the confusion that persists. With this review we would like to summarize the findings  and trigger further research. 

Round 2

Reviewer 3 Report

Overall, the article is of very low clinical value.